# REGULARIZING NEURAL NETWORKS BY PENALIZING CONFIDENT OUTPUT DISTRIBUTIONS

**Gabriel Pereyra** [*][†]
Google Brain
pereyra@google.com

**George Tucker** [*][†]
Google Brain
gjt@google.com

**Jan Chorowski**
Google Brain
chorowski@google.com

**Łukasz Kaiser**
Google Brain
lukaszkaiser@google.com

**Geoffrey Hinton**
University of Toronto & Google Brain
geoffhinton@google.com

## ABSTRACT

We systematically explore regularizing neural networks by penalizing low entropy output distributions. We show that penalizing low entropy output distributions, which has been shown to improve exploration in reinforcement learning, acts as a strong regularizer in supervised learning. Furthermore, we connect a maximum entropy based confidence penalty to label smoothing through the direction of the KL divergence. We exhaustively evaluate the proposed confidence penalty and label smoothing on 6 common benchmarks: image classification (MNIST and Cifar-10), language modeling (Penn Treebank), machine translation (WMT'14 English-to-German), and speech recognition (TIMIT and WSJ). We find that both label smoothing and the confidence penalty improve state-of-the-art models across benchmarks without modifying existing hyperparameters, suggesting the wide applicability of these regularizers.

## 1 INTRODUCTION

Large neural networks with millions of parameters achieve strong performance on image classification (Szegedy et al., 2015a), machine translation (Wu et al., 2016), language modeling (Jozefowicz et al., 2016), and speech recognition (Graves et al., 2013). However, despite using large datasets, neural networks are still prone to overfitting. Numerous techniques have been proposed to prevent overfitting, including early stopping, L1/L2 regularization (weight decay), dropout (Srivastava et al., 2014), and batch normalization (Ioffe & Szegedy, 2015). These techniques, along with most other forms of regularization, act on the hidden activations or weights of a neural network. Alternatively, regularizing the output distribution of large, deep neural networks has largely been unexplored.

To motivate output regularizers, we can view the knowledge of a model as the conditional distribution it produces over outputs given an input (Hinton et al., 2015) as opposed to the learned values of its parameters. Given this functional view of knowledge, the probabilities assigned to class labels that are incorrect (according to the training data) are part of the knowledge of the network. For example, when shown an image of a BMW, a network that assigns a probability of $10^{-3}$ to "Audi" and $10^{-9}$ to "carrot" is clearly better than a network that assigns $10^{-9}$ to "Audi" and $10^{-3}$ to carrot, all else being equal. One reason it is better is that the probabilities assigned to incorrect classes are an indication of how the network generalizes. Distillation (Hinton et al., 2015; Bucilu et al., 2006) exploits this fact by explicitly training a small network to assign the same probabilities to incorrect classes as a large network or ensemble of networks that generalizes well. Further, by operating on the output distribution that has a natural scale rather than on internal weights, whose significance depends on the values of the other weights, output regularization has the property that it is invariant to the parameterization of the underlying neural network.

---

[*]Work done as part of the Google Brain Residency Program
[†]Equal Contribution

In this paper, we systematically evaluated two output regularizers: a maximum entropy based confidence penalty and label smoothing (uniform and unigram) for large, deep neural networks on 6 common benchmarks: image classification (MNIST and Cifar-10), language modeling (Penn Treebank), machine translation (WMT'14 English-to-German), and speech recognition (TIMIT and WSJ). We find that both label smoothing and the confidence penalty improve state-of-the-art models across benchmarks without modifying existing hyperparameters.

## 2 RELATED WORK

The maximum entropy principle (Jaynes, 1957) has a long history with deep connections to many areas of machine learning including unsupervised learning, supervised learning, and reinforcement learning. In supervised learning, we can search for the model with maximum entropy subject to constraints on empirical statistics, which naturally gives rise to maximum likelihood in log-linear models (see (Berger et al., 1996) for a review). Deterministic annealing Rose (1998) is a general approach for optimization that is widely applicable, avoids local minima, and can minimize discrete objectives, and it can be derived from the maximum entropy principle. Closely related to our work, Miller et al. (1996) apply deterministic annealing to train multilayer perceptrons, where an entropy based regularizer is introduced and slowly annealed. However, their focus is avoiding poor initialization and local minima, and while they find that deterministic annealing helps, the improvement diminishes quickly as the number of hidden units exceeds eight.

In reinforcement learning, encouraging the policy to have an output distribution with high entropy has been used to improve exploration (Williams & Peng, 1991). This prevents the policy from converging early and leads to improved performance (Mnih et al., 2016). Penalizing low entropy has also been used when combining reinforcement learning and supervised learning to train a neural speech recognition model to learn when to emit tokens (Luo et al., 2016). When learning to emit, the entropy of the emission policy was added to the training objective and was annealed throughout training. Indeed, in recent work on reward augmented maximum likelihood (Norouzi et al., 2016), this entropy augmented reinforcement learning objective played a direct role in linking maximum likelihood and reinforcement learning objectives.

Penalizing the entropy of a network's output distribution has not been evaluated for large deep neural networks in supervised learning, but a closely related idea, label smoothing regularization, has been shown to improve generalization (Szegedy et al., 2015b). Label smoothing regularization estimates the marginalized effect of label-dropout during training, reducing overfitting by preventing a network from assigning full probability to each training example and maintaining a reasonable ratio between the logits of the incorrect classes. Simply adding label noise has also been shown to be effective at regularizing neural networks (Xie et al., 2016). Instead of smoothing the labels with a uniform distribution, as in label smoothing, we can smooth the labels with a teacher model (Hinton et al., 2015) or the model's own distribution (Reed et al., 2014). Distillation and self-distillation both regularize a network by incorporating information about the ratios between incorrect classes.

Virtual adversarial training (VAT) (Miyato et al., 2015) is another promising smoothing regularizer. However, we did not compare to VAT because it has multiple hyperparameters and the approximated gradient of the local distributional smoothness can be computed with no more than three pairs of forward and back propagations, which is significantly more computation in grid-searching and training than the other approaches we compared to.

## 3 DIRECTLY PENALIZING CONFIDENCE

Confident predictions correspond to output distributions that have low entropy. A network is overconfident when it places all probability on a single class in the training set, which is often a symptom of overfitting (Szegedy et al., 2015b). The confidence penalty constitutes a regularization term that prevents these peaked distributions, leading to better generalization.

A neural network produces a conditional distribution $p_\theta(\boldsymbol{y}|\boldsymbol{x})$ over classes $\boldsymbol{y}$ given an input $\boldsymbol{x}$ through a softmax function. The entropy of this conditional distribution is given by

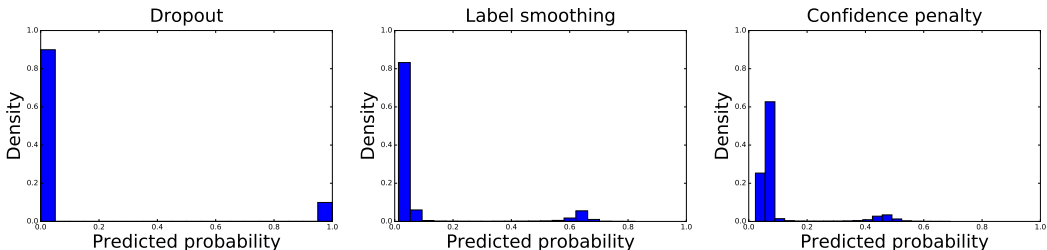

Figure 1: Distribution of the magnitude of softmax probabilities on the MNIST validation set. A fully-connected, 2-layer, 1024-unit neural network was trained with dropout (left), label smoothing (center), and the confidence penalty (right). Dropout leads to a softmax distribution where probabilities are either 0 or 1. By contrast, both label smoothing and the confidence penalty lead to smoother output distributions, which results in better generalization.

$$H(p_\theta(\boldsymbol{y}|\boldsymbol{x})) = -\sum_i p_\theta(\boldsymbol{y}_i|\boldsymbol{x}) \log(p_\theta(\boldsymbol{y}_i|\boldsymbol{x})).$$

To penalize confident output distributions, we add the negative entropy to the negative log-likelihood during training

$$\mathcal{L}(\theta) = -\sum \log p_\theta(\boldsymbol{y}|\boldsymbol{x}) - \beta H(p_\theta(\boldsymbol{y}|\boldsymbol{x})),$$

where $\beta$ controls the strength of the confidence penalty. Notably, the gradient of the entropy term with respect to the logits is simple to compute. Denoting the $i$th logit by $\boldsymbol{z}_i$, then

$$\frac{\partial H(p_\theta)}{\partial \boldsymbol{z}_i} = p_\theta(\boldsymbol{y}_i|\boldsymbol{x}) \left( -\log p_\theta(\boldsymbol{y}_i|\boldsymbol{x}) - H(p_\theta) \right),$$

which is the weighted deviation from the mean.

## 3.1 Annealing and Thresholding the Confidence Penalty

In reinforcement learning, penalizing low entropy distributions prevents a policy network from converging early and encourages exploration. However, in supervised learning, we typically want quick convergence, while preventing overfitting near the end of training, suggesting a confidence penalty that is weak at the beginning of training and strong near convergence. A simple way to achieve this is to anneal the confidence penalty.

Another way to strengthen the confidence penalty as training progresses is to only penalize output distributions when they are below a certain entropy threshold. We can achieve this by adding a hinge loss to the confidence penalty, leading to an objective of the form

$$\mathcal{L}(\theta) = -\sum \log p_\theta(\boldsymbol{y}|\boldsymbol{x}) - \beta \max(0, \Gamma - H(p_\theta(\boldsymbol{y}|\boldsymbol{x})),$$

where $\Gamma$ is the entropy threshold below which we begin applying the confidence penalty.

Initial experiments suggest that thresholding the confidence penalty leads to faster convergence at the cost of introducing an extra hyper-parameter. For the majority of our experiments, we were able to achieve comparable performance without using the thresholded version. For the sake of simplicity, we focus on the single hyper-parameter version in our experiments.

## 3.2 CONNECTION TO LABEL SMOOTHING

Label smoothing estimates the marginalized effect of label noise during training. When the prior label distribution is uniform, label smoothing is equivalent to adding the KL divergence between the uniform distribution $u$ and the network's predicted distribution $p_\theta$ to the negative log-likelihood

$$\mathcal{L}(\theta) = -\sum \log p_\theta(\boldsymbol{y}|\boldsymbol{x}) - D_{KL}(u\|p_\theta(\boldsymbol{y}|\boldsymbol{x})).$$

By reversing the direction of the KL divergence, $D_{KL}(p_\theta(\boldsymbol{y}|\boldsymbol{x})\|u)$, we recover the confidence penalty. This interpretation suggests further confidence regularizers that use alternative target distributions instead of the uniform distribution. We leave the exploration of these regularizers to future work.

## 4 EXPERIMENTS

We evaluated the confidence penalty and label smoothing on MNIST and CIFAR-10 for image classification, Penn Treebank for language modeling, WMT'14 English-to-German for machine translation, and TIMIT and WSJ for speech recognition. All models were implemented using TensorFlow (Abadi et al., 2016) and trained on NVIDIA Tesla K40 or K80 GPUs.

### 4.1 IMAGE CLASSIFICATION

#### 4.1.1 MNIST

As a preliminary experiment, we evaluated the approaches on the standard MNIST digit recognition task. We used the standard split into 60k training images and 10k testing images. We use the last 10k images of the training set as a held-out validation set for hyper-parameter tuning and then retrained the models on the entire dataset with the best configuration.

We trained fully-connected, ReLu activation neural networks with 1024 units per layer and two hidden layers. Weights were initialized from a normal distribution with standard deviation 0.01. Models were optimized with stochastic gradient descent with a constant learning rate 0.05 (except for dropout where we set the learning rate to 0.001).

For label smoothing, we varied the smoothing parameter in the range [0.05, 0.1, 0.2, 0.3, 0.4, 0.5], and found 0.1 to work best for both methods. For the confidence penalty, we varied the weight values over [0.1, 0.3, 0.5, 1.0, 2.0, 4.0, 8.0] and found a confidence penalty weight of 1.0 to work best.

We also plotted the norm of the gradient as training progressed in Figure 2. We observed that label smoothing and confidence penalty had smaller gradient norms and converged more quickly than models regularized with dropout. If the output distributions is peaked on a misclassified example, the model receives a large gradient. This may explain why the regularized models have smaller gradient norms.

| Model | Layers | Size | Test |
|---|---|---|---|
| Wan et al. (2013) - Unregularized | 2 | 800 | 1.40% |
| Srivastava et al. (2014) - Dropout | 3 | 1024 | 1.25% |
| Wan et al. (2013) - DropConnect | 2 | 800 | 1.20% |
| Srivastava et al. (2014) - MaxNorm + Dropout | 2 | 8192 | **0.95%** |
| Dropout | 2 | 1024 | $1.28 \pm 0.06\%$ |
| Label Smoothing | 2 | 1024 | $1.23 \pm 0.06\%$ |
| Confidence Penalty | 2 | 1024 | $\mathbf{1.17 \pm 0.06\%}$ |

Table 1: Test error (%) for permutation-invariant MNIST.

### 4.1.2   CIFAR-10

CIFAR-10 is an image classification dataset consisting of 32x32x3 RGB images of 10 classes. The dataset is split into 50k training images and 10k testing images. We use the last 5k images of the training set as a held-out validation set for hyper-parameter tuning, as is common practice.

For our experiments, we used a densely connected convolutional neural network, which represents the current state-of-the-art on CIFAR-10 (Huang et al., 2016a). We use the small configuration from (Huang et al., 2016a), which consists of 40-layers, with a growth rate of 12. All models were trained for 300 epochs, with a batch-size of 50 and a learning rate 0.1. The learning rate was reduced by a factor of 10 at 150 and 225 epochs. We present results for training without data-augmentation. We found that the confidence penalty did not lead to improved performance when training with data augmentation, however neither did other regularization techniques, including dropout.

For our final test scores, we trained on the entire training set. For label smoothing, we tried smoothing parameter values of [0.05, 0.1, 0.2, 0.3, 0.4, 0.5], and found 0.1 to work best. For the confidence penalty, we performed a grid search over confidence penalty weight values of [0.1, 0.25, 0.5, 1.0, 1.5] and found a confidence penalty weight of 0.1 to work best.

| Model | Layers | Parameters | Test |
|---|---|---|---|
| He et al. (2015) - Residual CNN | 110 | 1.7M | 13.63% |
| Huang et al. (2016b) - Stochastic Depth Residual CNN | 110 | 1.7M | 11.66% |
| Larsson et al. (2016) - Fractal CNN | 21 | 38.6M | 10.18% |
| Larsson et al. (2016) - Fractal CNN (Dropout) | 21 | 38.6M | 7.33% |
| Huang et al. (2016a) - Densely Connected CNN | 40 | 1.0M | 7.00% |
| Huang et al. (2016a) - Densely Connected CNN | 100 | 7.0M | **5.77%** |
| Densely Connected CNN (Dropout) | 40 | 1.0M | 7.04% |
| Densely Connected CNN (Dropout + Label Smoothing) | 40 | 1.0M | 6.89% |
| Densely Connected CNN (Dropout + Confidence Penalty) | 40 | 1.0M | **6.77%** |

Table 2: Test error (%) on Cifar-10 without data augmentation.

### 4.2   LANGUAGE MODELING

For language modeling, we found that confidence penalty significantly outperforms label noise and label smoothing. We performed word-level language modeling experiments using the Penn Tree-bank dataset (PTB) (Marcus et al., 1993). We used the hyper-parameter settings from the large configuration in (Zaremba et al., 2014). Briefly, we used a 2-layer, 1500-unit LSTM, with 65% dropout applied on all non-recurrent connections. We trained using stochastic gradient descent for 55 epochs, decaying the learning rate by 1.15 after 14 epochs, and clipped the norm of the gradients when they were larger than 10.

For label noise and label smoothing, we performed a grid search over noise and smoothing values of $[0.05, 0.1, 0.15, 0.2, 0.3, 0.4, 0.5]$. For label noise, we found $0.1$ to work best. For label smoothing, we found $0.1$ to work best. For the confidence penalty, we performed a grid search over confidence penalty weight values of $[0.1, 0.5, 1.0, 2.0, 3.0]$. We found a confidence penalty weight of $2.0$ to work best, which led to an improvement of 3.7 perplexity points over the baseline.

For reference, we also include results of the existing state-of-the-art models for the word-level language modeling task on PTB. Variational dropout (Gal, 2015) applies a fixed dropout mask (stochastic for each sample) at each time-step, instead of resampling at each time-step as in traditional dropout. Note, that we do not include the variational dropout results that use Monte Carlo (MC) model averaging, which achieves lower perplexity on the test set but requires 1000 model evaluations, which are then averaged. Recurrent highway networks (Zilly et al., 2016) currently represent the state-of-the-art performance on PTB.

| Model | Parameters | Validation | Test |
|---|---|---|---|
| Zaremba et al. (2014) - Regularized LSTM | 66M | 82.2 | 78.4 |
| Gal (2015) - Variational LSTM | 66M | 77.9 | 75.2 |
| Press & Wolf (2016) - Tied Variational LSTM | 51M | 79.6 | 75.0 |
| Merity et al. (2016) - Pointer Sentinel LSTM | 21M | 72.4 | 70.9 |
| Zilly et al. (2016) - Variational RHN | 32M | 71.2 | 68.5 |
| Zilly et al. (2016) - Tied Variational RHN | 24M | 68.1 | **66.0** |
| Regularized LSTM (label noise) | 66M | 79.7 | 77.7 |
| Regularized LSTM (label smoothing) | 66M | 78.9 | 76.6 |
| Regularized LSTM (unigram smoothing) | 66M | 79.1 | 76.3 |
| Regularized LSTM (confidence penalty) | 66M | 77.8 | **74.7** |

Table 3: Validation and test perplexity for word-level Penn Treebank.

### 4.3 MACHINE TRANSLATION

For machine translation, we evaluated the confidence penalty on the WMT'14 English-to-German translation task using Google's production-level translation system Wu et al. (2016). The training set consists of 5M sentence pairs, and we used newstest2012 and newtests2013 for validation and newstest2014 for testing. We report tokenized BLEU scores as computed by the `multi-bleu.perl` script from the Moses translation machine translation package.

Our model was an 8-layer sequence-to-sequence model with attention (Bahdanau et al., 2014). The first encoder was a bidirectional LSTM, the remaining encoder and decoder layers were unidirectional LSTMs, and the attention network was a single layer feed-forward network. Each layer had 512 units (compared to 1024 in (Wu et al., 2016)). The model was trained using 12 replicas running concurrently with asynchronous updates. Dropout of 30% was applied as described in (Zaremba et al., 2014). Optimization used a mix of Adam and SGD with gradient clipping. Unlike (Wu et al., 2016), we did not use reinforcement learning to fine-tune our model. We used a beam size of 12 during decoding. For more details, see (Wu et al., 2016).

For label smoothing, we performed a grid search over values $[0.05, 0.1, 0.2, 0.3, 0.4, 0.5]$ and found 0.1 to work best for both uniform and unigram smoothing. For the confidence penalty, we searched over values of $[0.5, 2.5, 4.5]$ and found a value of 2.5 to work best . For machine translation, we found label smoothing slightly outperformed confidence penalty. When applied without dropout, both lead to an improvement of just over 1 BLEU point (dropout leads to an improvement of just over 2 BLEU points). However, when combined with dropout, the effect of both regularizers was diminished.

| Model | Parameters | Validation | Test |
|---|---|---|---|
| Buck et al. (2014) - PBMT | - | - | 20.7 |
| Cho et al. (2015) - RNNSearch | - | - | 16.9 |
| Zhou et al. (2016) - Deep-Att | - | - | 20.6 |
| Luong et al. (2015) - P-Attention | 164M | - | 20.9 |
| Wu et al. (2016) - WPM-16K | 167M | - | 24.4 |
| Wu et al. (2016) - WPM-32K | 278M | - | **24.6** |
| WPM-32K (without dropout) | 94M | 22.33 | 21.24 |
| WPM-32K (label smoothing) | 94M | 23.85 | 22.42 |
| WPM-32K (confidence penalty) | 94M | 23.25 | 22.52 |
| WPM-32K (dropout) | 94M | $24.1 \pm 0.1$ | $23.41 \pm 0.04$ |
| WPM-32K (dropout + label smoothing) | 94M | $24.3 \pm 0.1$ | $\mathbf{23.52 \pm 0.03}$ |
| WPM-32K (dropout + unigram smoothing) | 94M | $24.3 \pm 0.1$ | $\mathbf{23.57 \pm 0.02}$ |
| WPM-32K (dropout + confidence penalty) | 94M | $24.3 \pm 0.1$ | $23.4 \pm 0.1$ |

Table 4: Validation and test BLEU for WMT'14 English-to-German. For the last four model configurations, we report the mean and standard error of the mean (SEM) over 5 random initializations.

### 4.4 SPEECH RECOGNITION

#### 4.4.1 TIMIT

In the TIMIT corpus, the training set consists of 3512 utterances, the validation set consists of 184 utterances and the test set consists of 192 utterances. All 61 phonemes were used during training and decoding, and during scoring, these 61 phonemes were reduced to 39 to compute the phoneme error rate (PER).

As our base model, we used a sequence-to-sequence model with attention. The encoder consisted of 3 bidirectional LSTM layers, the decoder consisted of a single unidirectional LSTM layer, and the attention network consisted of a single layer feed-forward network. All layers consisted of 256 units. Dropout of 15% was applied as described in Zaremba et al. (2014). We trained the model with asynchronous SGD with 5 replicas. We used a batch size of 32, a learning rate of 0.01, and momentum of 0.9. Gradients were clipped at 5.0. For more details, see Norouzi et al. (2016).

For label smoothing, we performed a grid search over values $[0.05, 0.1, 0.15, 0.2, 0.3, 0.4, 0.5, 0.6, 0.8]$ and found 0.2 to work best. For the confidence penalty, we performed a grid search over values of $[0.125, 0.25, 0.5, 1.0, 2.0, 4.0, 8.0]$ and found a value of 1.0 to work best. Label smoothing led to an absolute improvement over the dropout baseline of 1.6%, while the confidence penalty led to an absolute improvement of 1.2%.

| Model | Parameters | Validation | Test |
|---|---|---|---|
| Mohamed et al. (2012) - DNN-HMM | - | - | 20.7 |
| Norouzi et al. (2016) - RML | 6.5M | 18.0 | 19.9 |
| Graves et al. (2006) - CTC | 6.8M | - | 18.4 |
| Graves et al. (2013) - RNN Transducer | 4.3M | - | 17.7 |
| Tóth (2014) - CNN | - | 13.9 | **16.7** |
| Dropout | 6.5M | $21.0 \pm 0.1$ | $23.2 \pm 0.4$ |
| Dropout + Label Smoothing | 6.5M | $19.3 \pm 0.1$ | $\mathbf{21.6 \pm 0.2}$ |
| Dropout + Confidence Penalty | 6.5M | $19.9 \pm 0.2$ | $22.0 \pm 0.4$ |

Table 5: Validation and test phoneme error rates (PER) for TIMIT. We report the mean and SEM over 5 random initializations.

#### 4.4.2 WALL STREET JOURNAL

For the WSJ corpus we used attention-based sequence-to-sequence networks that directly predicted characters. We used the SI284 subset for training, DEV93 for validation, and EVAL92 for testing. We used 240-dimensional vectors consisting of 80-bin filterbank features augmented with their deltas and delta-deltas with per-speaker normalized mean and variances computed with Kaldi Povey et al. (2011). We did not use text-only data or separate language models during decoding.

Network architecture details were as follows. The encoder of the network consisted of 4 bidirectional LSTM layers each having 256 units, interleaved with 3 time-subsampling layers, configured to drop every second frame (Bahdanau et al., 2016; Chan et al., 2015). The decoder used a single LSTM layer with 256 units. The attention vectors were computed with a single layer feedforward network having 64 hidden units and the convolutional filters as described in Chorowski et al. (2015). Weights were initialized from a uniform distribution $[-0.075, 0.075]$. All models used weight decay of $10^{-6}$, additive Gaussian weight noise with standard deviation $0.075$, applied after 20K steps, and were trained for 650K steps. We used the ADAM optimizer asynchronously over 8 GPUs. We used a learning rate of $10^{-3}$, which was reduced to $10^{-4}$ after 400K and $10^{-5}$ after 500K steps.

We tested three methods of increasing the entropy of outputs: the confidence penalty and two variants of label smoothing: uniform and unigram. All resulted in improved Word Error Rates (WER), however the unigram smoothing resulted in the greatest WER reduction, and we found it to be least sensitive to its hyperparameter (the smoothing value). Furthermore, uniform smoothing and confidence penalty required masking network outputs corresponding to tokens that never appeared as labels, such as the start-of-sequence token.

Table 6 compares the performance of the regularized networks with several recent results. We observe that the benefits of label smoothing (WER reduction from 14.2 to 11) improve over the recently proposed Latent Sequence Decompositions (LSD) method (Chan et al., 2016) which reduces the WER from 14.7 to 12.9 by extending the space of output tokens to dynamically chosen character n-grams.

| Model | Parameters | Validation | Test |
|---|---|---|---|
| Graves & Jaitly (2014) - CTC | 26.5M | - | 27.3 |
| Bahdanau et al. (2016) - seq2seq | 5.7M | - | 18.6 |
| Chan et al. (2016) - Baseline | 5.1M | - | 14.7 |
| Chan et al. (2016) - LSD | 5.9M | - | 12.9 |
| Baseline | 6.6M | 17.9 | 14.2 |
| Uniform Label Smoothing | 6.6M | 14.7 | 11.3 |
| Unigram Label Smoothing | 6.6M | $14.0 \pm 0.25$ | $\mathbf{11.0 \pm 0.35}$ |
| Confidence Penalty | 6.6M | 17.2 | 12.7 |

Table 6: Validation and test word error rates (WER) for WSJ. For Baseline, Uniform Label Smoothing and Confidence Penalty we report the average over two runs. For the best setting (Unigram Label Smoothing), we report the average over 6 runs together with the standard deviation.

## 5 CONCLUSION

Motivated by recent successes of output regularizers (Szegedy et al., 2015b; Xie et al., 2016), we conduct a systematic evaluation of two output regularizers: the confidence penalty and label smoothing. We show that this form of regularization, which has been shown to improve exploration in reinforcement learning, also acts as a strong regularizer in supervised learning. We find that both the confidence penalty and label smoothing improve a wide range of state-of-the-art models, without the need to modify hyper-parameters.

ACKNOWLEDGMENTS

We would like to thank Sergey Ioffe, Alex Alemi and Navdeep Jaitly for helpful discussions. We would also like to thank Prajit Ramachandran, Barret Zoph, Mohammad Norouzi, and Yonghui Wu for technical help with the various models used in our experiments. We thank the anonymous reviewers for insightful comments.

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

APPENDIX

## 6 GRADIENT NORMS

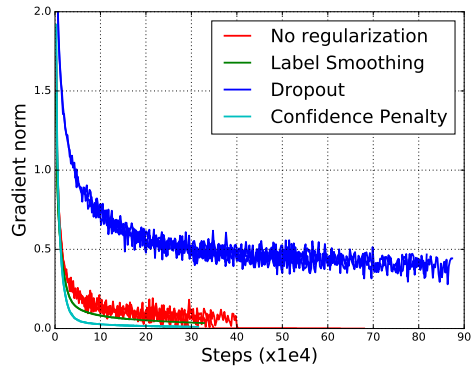

Figure 2: Norm of the gradient as training proceeds on the MNIST dataset. We plot the norm of the gradient while training with confidence penalty, dropout, label smoothing, and without regularization. We use early stopping on the validation set, which explains the difference in training steps between methods. Both confidence penalty and label smoothing result in smaller gradient norm.

