# Peer review of "Regularizing Neural Networks by Penalizing Confident Output Distributions"

_ICLR 2017 — rejected_

[Official Review · AnonReviewer5 · rating 5 · confidence 4 · 15 Dec 2016]
**Simple idea, analysis lacking, comprehensive experiments**

The authors propose a simple idea. They penalize confident predictions by using the entropy of the predictive distribution as a regularizer. The authors consider two variations on this idea. In one, they penalize the divergence from the uniform distribution. In the other variation, they penalize distance from the base rates. They term this variation "unigram" but I find the name odd as I've never seen multi-class labels described as unigrams before. What would a bigram be? 

The idea is simple,  and while it's been used in the context of reinforcement learning, it hasn't been popularized as a regularizer for improving generalization in supervised learning. 

The justifications for the idea still lacks analysis. And the author responses comparing it to L2 regularization have some holes. A simple number line example with polynomial regression makes clear how L2 regularization could prevent a model from badly overfitting to accommodate every data point. In contrast, it seems trivial to fit every data point and satisfy arbitrarily high entropy. Of course, the un-regularized optimization is to maximize log likelihood, not simply to maximize accuracy.  And perhaps something interesting may be happening at the interplay between the log likelihood objective and the regularization objective. But the paper doesn't indicate precisely what.

I could imagine the following scenario: when the network outputs probabilities near 0, it can get high loss (if the label is 1). The entropy regularization could be stabilizing the gradient, preventing sharp loss on outlier examples. The regularization then might owe mainly to faster convergence. Could the authors analyze the effect empirically, on the distribution of the gradient norms? 

The strength of this paper is its empirical rigor. The authors take their idea and put it through its paces on a host of popular and classic benchmarks spanning CNNs and RNNs. It appears that on some datasets, especially language modeling, the confidence penalty outperforms label smoothing. 

At present, I rate this paper as a borderline contribution but I'm open to revising my review pending further modifications. 

Typo:
In related work: "Penalizing entropy" - you mean penalizing low entropy

[Official Review · AnonReviewer4 · rating 5 · confidence 4 · 17 Dec 2016]
**perfectly sensible idea, with apparently good results, but lacks scholarship**

Specifically, this paper suggests regularizing the estimator of a probability distribution to prefer high-entropy distributions.  This avoids overfitting.

I generally like this idea.  Regularizing the behavior of the model often makes more sense than regularizing its parameters.  After all, the behavior is interpretable, whereas the parameters are uninterpretable and work together in mysterious ways to produce the behavior.  So one might be able to choose a more sensible prior over the behavior.  In other words, prefer parameters not because they are individually close to 0 but because they jointly lead to a distribution that is plausible or low-risk a priori.

Pro: I believe that the idea is natural and sound (that is, I do not share the doubts of AnonReviewer5).

Pro: It's possible that this hasn't been well-explored yet in neural networks (not sure).

Pro: The experimental results look good.  So maybe everyone should use this kind of regularizer. 

Con: It is a kind of pollution of the scientific literature to introduce this idea to the community as if it were unconnected to (almost) anything else in machine learning.  There are many, many papers that include a scaled entropy term in the optimization objective!  It's not just for reinforcement learning.  Please see the long list of connections in my pre-review questions / comments.  

Con: Experimental results should always be accompanied by significance tests and error analysis.  Is your trained model actually doing better on the distribution of test data, or was your test set too small to tell?  Are the improvements robust across many different training sets?  What errors does your model fix, and what errors does it introduce?  

Summary recommendation: Revise and resubmit.  ICLR has lots of submissions.  I would prefer to reward authors who not only tried something, but who properly contextualized it and carefully evaluated it.  Otherwise, there's a race to the bottom where everyone wants to be the first to try something, so that readers are confronted with a confusing sea of slapdash papers with unclear relationships.

[Official Review · AnonReviewer3 · rating 6 · confidence 4 · 19 Dec 2016]
**Experimental investigations on a slightly modified version of label smoothing.**

The paper experimentally investigates a slightly modified version of label smoothing technique for neural network training, and reports results on various tasks. Such smoothing idea is not new, but was not investigated previously in wide range of machine learning tasks.

Comments:
The paper should report the state-of-the-art results for speech recognition tasks (TIMIT, WSJ), even if models are not directly comparable.
The error back-propagation of label smoothing through softmax is straightforward and efficient. Is there an efficient solution for BP of the entropy smoothing through softmax?
Although the classification accuracy could remain the same, the model will not estimate the true posterior distribution with this kind of smoothing.
This might be an issue in complex machine learning problems where the decision is made on higher level and based on the posterior estimations, e.g. language models in speech recognition.
More motivation is necessary for the proposed smoothing.

[Final Decision · Program Chairs · 06 Feb 2017]
**ICLR committee final decision**

The reviewers agreed that the idea proposed in this paper is sensible and possibly very useful, and that the experiments are thorough with good results. However, they share strong doubts regarding the novelty of the proposed approach. Hopefully the discussion will help the authors refine this work. With a more thorough discussion of related ideas, such as entropy maximization, within the machine learning literature and a careful placement of this idea within that context, this could be a strong submission to a future conference.